# Metabolic Response After a Single Maximal Exercise Session in Physically Inactive Young Adults (EASY Study): Relevancy of Adiponectin Isoforms

**DOI:** 10.3390/biom15030314

**Published:** 2025-02-20

**Authors:** Johnattan Cano-Montoya, Amanda Bentes, Yanara Pavez, Paola Rubilar, Carolina Lavoz, Pamela Ehrenfeld, Viviana Sandoval, Sergio Martínez-Huenchullán

**Affiliations:** 1Carrera de Kinesiología, Facultad de Odontología y Ciencias de la Rehabilitación, Universidad San Sebastián, Valdivia 5090000, Chile; johnattan.cano@uss.cl; 2Instituto de Anatomía, Histología y Patología, Facultad de Medicina, Universidad Austral de Chile, Valdivia 5110566, Chile; amanda.bentes@alumnos.uach.cl (A.B.); ingrid.ehrenfeld@uach.cl (P.E.); 3Carrera de Tecnología Médica, Facultad de Medicina y Ciencia, Universidad San Sebastián, Valdivia 5090000, Chile; yanara.pavez@uss.cl (Y.P.); prubilars@docente.uss.cl (P.R.); 4Instituto de Medicina, Facultad de Medicina, Universidad Austral de Chile, Valdivia 5110566, Chile; carolina.lavoz@uach.cl; 5Centro Interdisciplinario de Estudios del Sistema Nervioso (CISNe), Universidad Austral de Chile, Valdivia 5110566, Chile; 6Carrera de Nutrición y Dietética, Facultad de Ciencias para el Cuidado de la Salud, Universidad San Sebastián, Valdivia 5090000, Chile

**Keywords:** exercise, adiponectin, physical inactivity

## Abstract

The metabolic response to a maximal exercise test in physically inactive adults remains poorly understood, particularly regarding the role of adiponectin, an adipokine with insulin-sensitizing and anti-inflammatory properties. Adiponectin circulates in three isoforms—low (LMW), medium (MMW), and high-molecular-weight (HMW)—with differing bioactivities. While exercise is known to influence adiponectin levels, evidence is conflicting, and few studies have explored isoform-specific changes. This study aimed to evaluate the effects of a single maximal exercise session on circulating adiponectin isoforms and their associations with metabolic and kidney function markers in physically inactive young adults. In this quasi-experimental study, twenty-one physically inactive participants (mean age 24.6 ± 2.1 years, 85.7% women) completed a progressive cycle ergometer test. Circulating levels of LMW and MMW adiponectin, metabolic outcomes (e.g., cholesterol, triglycerides, fibroblast growth factor 21 (FGF21)), and kidney function markers (e.g., creatinine, proteinuria) were assessed before and after exercise using biochemical assays and Western blotting. Comparisons between pre- and post-exercise values were made with the Wilcoxon test. Exercise increased lipid metabolism markers (total cholesterol, triglycerides, HDL) and kidney stress indicators (albuminuria, proteinuria) (*p* < 0.05). LMW and MMW adiponectin levels showed no significant overall changes, but LMW adiponectin positively correlated with changes in total cholesterol and FGF21, while MMW adiponectin negatively correlated with creatinine and proteinuria (*p* < 0.05). HMW adiponectin was undetectable by our methods. A single maximal exercise session revealed isoform-specific associations between adiponectin and metabolic or kidney stress markers, emphasizing the complex role of adiponectin in exercise-induced metabolic responses. Future research should explore mechanisms underlying these differential associations to optimize exercise interventions for metabolic health improvement.

## 1. Introduction

Physical inactivity is defined as failing to meet the minimum levels of spontaneous activity required to maintain a healthy lifestyle. The World Health Organization defines it as engaging in less than 150 min per week of moderate-intensity physical activity or less than 75 min per week of vigorous-intensity activity [1]. Maintaining a physically inactive lifestyle is associated with an increased risk of developing a range of cardiorespiratory and metabolic diseases, including hypertension, dyslipidemia, obesity, insulin resistance, and type 2 diabetes mellitus [2,3,4].

Given these risks, strategies to counter the effects of physical inactivity have gained increasing scientific interest, particularly concerning cardiometabolic health [5]. In this regard, the metabolic effects of physical exercise have been well-documented [6]. However, since the outcomes of exercise depend on its modality, frequency, duration, and intensity, these effects can vary significantly between different exercise prescriptions [7]. For example, when intensity is manipulated, vigorous exercises (where only a limited number of repetitions can be performed) primarily activate anaerobic pathways to extract energy in skeletal muscles, relying predominantly on glycogen. In contrast, exercises performed at a lower intensity that allow sustained effort for several minutes activate pathways dependent on nutrient oxidation, utilizing carbohydrates and lipids [8]. Given these distinctions, aerobic exercise is often highlighted in the context of obesity, particularly due to its beneficial effects on glucose transport and insulin sensitivity under both normal and pathological conditions (e.g., type 2 diabetes) [9].

In recent years, growing evidence has highlighted that exercise-induced metabolic responses of exercise are mediated by exerkines, which are signaling molecules released into circulation in response to exercise [10]. One of such exerkines is adiponectin, a 30 kDa protein primarily secreted by adipose tissue [11] and skeletal muscle [12]. Adiponectin has the ability to form complexes due to its carboxy-terminal globular domain and amino-terminal collagen domain [13,14]. These complexes are categorized into low (LMW), medium (MMW), and high-molecular-weight (HMW) isoforms [15], with increasing bioactivity [14] giving its effects through AdipoR1 and AdipoR2 receptors [16]. Adiponectin is known for its insulin-sensitizing and anti-inflammatory properties [17], making it a strong candidate for mediating exercise-induced metabolic benefits. Previous studies from our research group, focused on studying the metabolic benefits of exercise in an obesity context, have shown that 10 weeks of exercise training increase adiponectin levels in the skeletal muscles of mice fed a high-fat diet [18,19]; however, this increase was dependent on exercise intensity [20]. In a clinical context, we observed similar increases in circulating adiponectin levels after 4 weeks of exercise training in candidates for bariatric surgery [21], but again, these changes were intensity dependent.

More recently, we found that these effects can occur after a single session of moderate-intensity exercise [22]. However, given that the response appears to be sensitive to exercise intensity, it remains unclear whether these effects can be replicated under high-intensity conditions and, secondly, what the implications and associations of potential changes in adiponectin isoforms after exercise might be. Therefore, this study aimed to analyze the effects of a progressive exercise session on circulating adiponectin isoforms and investigate the associations of these potential changes with metabolic outcomes in physically inactive young adults.

## 2. Materials and Methods

### 2.1. Study Design and Ethical Considerations

This is a quasi-experimental, pre-post comparison study where the primary outcome was to compare the metabolic responses before and after a maximal physical effort during a progressive cycle ergometer test (Figure 1). The study was reviewed and approved by the Ethics Committee of the Los Ríos Health Service (code 222/2022), and all participants provided written informed consent before participating. This study is registered at ClinicalTrials.gov (NCT06680713).

### 2.2. Participants

This study included individuals aged 18 to 30 who were physically inactive (engaging in less than 150 min/week of moderate physical activity and/or 75 min/week of vigorous physical activity) and had no medical contraindications for physical activity. Participants taking medications that influenced blood glucose or insulin levels and/or had anti-inflammatory effects were excluded. Participants who were physically active (performed more than 150 min per week or 75 min per week of moderate or vigorous physical activity, respectively) were excluded prior to the start of the trial. Sample size calculation was based on the following parameters: 95% power, an alpha value of 0.05, a pre- and post-intervention mean difference of fibroblast growth factor 21 (FGF21) of 50 pg/mL [23] (e.g., 100 vs. 150 pg/mL), and a standard deviation of 50 pg/mL for both groups. Based on these data, the sample size required was 16 participants; with a 20% drop-out rate, the final sample size was set at 20 participants, recruited by convenience. FGF21 was considered for the sample size calculation, considering its role in modulating glucose and lipid metabolism, along with previously reported effects of exercise on the circulatory levels of this protein [18,22].

### 2.3. Phenotypic Measurements and Maximal Effort Test

Participants’ anthropometry was assessed using weight, height, body mass index (BMI), waist circumference, and waist-to-hip and waist-to-height ratios. The percentage of body fat and muscle mass relative to body weight was measured using bioimpedance (InBody^®^ 270, InBody Co., Ltd., Seoul, Korea). Total physical activity levels were evaluated using the International Physical Activity Questionnaire (IPAQ), validated for the national population [24]. The values reported by this instrument represent the total physical activity levels, which include light, moderate, and vigorous activities across different domains (work, transport, domestic, and leisure). After 5 min of seated rest, heart rate (HR) was measured with a Polar^®^ H10 (Kempele, Finland) heart rate monitor, blood pressure (BP) with an OMRON^®^ (Kyoto, Japan) sphygmomanometer, oxygen saturation (SatO_2_) with a Nonin^®^ Onyx Vantage device (Plymouth, USA), and the ratings of perceived exertion (RPE) using the original Borg scale. Additionally, muscle oximetry (SmO_2_) was assessed with a wireless oximeter (Moxy^®^ 5, Minnesota, USA) placed on the belly of the right quadriceps vastus lateralis muscle. The maximal effort test consisted of an incremental load test on a cycle ergometer or stationary bike, where the pedaling load was increased by 30 W every 3 min, starting at 30 W and continuing until fatigue, maintaining a pedaling cadence of 60 revolutions per minute (RPM). During the test, HR, SatO_2_, SmO_2_, and RPE were monitored every minute, while BP was measured at the end of the test. Test termination criteria included achieving 85% or more of maximum HR, BP exceeding 180/100 mmHg, RPE of 17/20 or higher, and/or voluntary fatigue or inability to maintain pedaling cadence. Immediately after the test, HR, BP, SatO_2_, SmO_2_, and RPE were reassessed.

### 2.4. Laboratory Tests

Before the test, blood lactate levels were measured via a finger prick using a specific reactive strip and an AccuTrend Plus^®^ device(Basel, Switzerland). Venous blood samples were collected from the antecubital vein in a 2 mL EDTA tube and another tube without anticoagulant (with separator gel) for subsequent plasma measurements of fibroblast growth factor 21 (FGF21) using a specific ELISA kit (RayBio^®^, catalog ELH-FGF21, Georgia, USA) and biochemical tests such as blood glucose, insulin, and the HOMA-IR index. After the test, additional venous blood samples and post-exercise blood lactate levels were collected. This method was chosen because lactate concentrations are highly dynamic post-exercise, with a reported 50% drop within 4 min in healthy individuals [25]. In terms of urine sample collection, participants were instructed to collect it during the first urination in the morning of the same day of the exercise session, whereas the post-exercise urine samples were collected immediately after the test. The outcomes measured in urine were albuminuria, creatinine clearance, albuminuria/creatinine clearance ratio, glucosuria, and proteinuria. These tests were conducted at the Universidad Austral de Chile Clinical Laboratory following standard protocols.

### 2.5. Adiponectin and Western Immunoblotting

To measure circulating adiponectin, the Western immunoblotting technique was used according to the following protocol: 0.5 µL of plasma was mixed with 2.5 µL of 6× loading buffer and 12 µL of RIPA buffer and loaded onto an acrylamide gradient gel (Bio-Rad^®^, catalog number 1704158, Hercules, USA). After the proteins were separated by electrophoresis (80–120 W for approximately 60 min), they were transferred onto a nitrocellulose membrane (Bio-Rad^®^, catalog number 1704158) using a semi-dry transfer system (Trans-Blot^®^ Turbo™ Transfer System, Bio-Rad^®^, Hercules, USA). The membranes were blocked for one hour with 5% bovine serum albumin diluted in TBST buffer. After washing in TBST, they were incubated overnight at 4 °C with a primary antibody against human adiponectin (1:1000 dilution, rabbit monoclonal antibody, Cell Signaling^®^, catalog number 2789S, Danvers, USA). Subsequently, the membranes were washed in TBST and incubated with a secondary antibody conjugated to peroxidase (1:2500 dilution, anti-rabbit IgG, Cell Signaling^®^, catalog number 7074S) for one hour at room temperature. Finally, the membranes were washed in TBST and developed with a chemiluminescent substrate (ClarityTM Western ECL substrate, Bio-Rad^®^, catalog number 170-5061) and visualized using a densitometric system (G:Box system, Syngene^®^, USA). Band quantification was performed using the ImageJ software Version 1.53t, and total protein loading was quantified by staining the entire membrane with Ponceau red, where the low (LMW), medium (MMW), and high-molecular-weight (HMW) adiponectin isoforms were measured.

### 2.6. Statistical Analysis

Quantitative variables were summarized as means and standard deviations, while qualitative variables were expressed as absolute frequencies. Data distribution normality was assessed using the Kolmogorov–Smirnov test. Pre- and post-training comparisons of variables of interest were conducted using the Wilcoxon test. Bivariable correlations were calculated using Spearman’s rho index. A *p*-value of less than 0.05 was considered statistically significant for all analyses. SPSS version 20 and GraphPad version 8 software were used for the analyses.

## 3. Results

Of the 22 participants recruited, 21 were included in the analysis. One participant was excluded due to difficulty in obtaining a post-exercise blood sample because of challenging venous access (Figure 2). The sample consisted primarily of women, with a BMI indicative of being overweight, a muscle mass accounting for 35% of total body weight, and a median of sedentary behavior of five hours per day of sitting time (Table 1).

The exercise session elicited physiological responses indicative of maximal effort, as evidenced by participants using 90% of their heart rate (HR) reserve. Additionally, significant increases were observed in systolic blood pressure, perceived exertion (RPE), and lactate levels (Table 2; all *p* < 0.05). Notably, all participants completed the exercise session safely, with only one reporting mild dizziness at the end, which resolved spontaneously within five minutes.

Regarding metabolic responses, participants demonstrated significant increases in several lipid profile parameters after exercise, including total cholesterol, triglycerides, HDL, LDL, and non-HDL cholesterol (Table 3; all *p* < 0.05). Furthermore, markers of kidney function also showed increases post-exercise, particularly albuminuria, the albuminuria/creatinine clearance ratio, and proteinuria (Table 4; all *p* < 0.05).

The low molecular weight (LMW) and medium molecular weight (MMW) adiponectin isoforms did not show significant changes after the single exercise session (Figure 3A,B; *p* > 0.05). However, significant positive correlations were observed between the change from baseline (∆) in LMW adiponectin and the ∆ in total cholesterol and ∆ in FGF21 (Figure 3C,D). Conversely, ∆ in MMW adiponectin was negatively correlated with ∆ in creatinine and ∆ in proteinuria levels (Figure 3E,F). It is worth mentioning that the HMW adiponectin isoform was unobservable through our methods (below our detection range).

## 4. Discussion

This study aimed to analyze the effects of a progressive exercise session on circulating adiponectin isoforms and to investigate the associations of these potential changes with metabolic outcomes in physically inactive young adults. The main findings revealed that this type of exercise does not alter LMW and MMW isoforms of adiponectin, whereas increases in circulating lipids and markers of kidney function were observed. Interestingly, due to the variability in adiponectin responses to exercise, we found that changes in the LMW isoform after exercise were positively associated with changes in total cholesterol and FGF21. Conversely, changes in MMW adiponectin were negatively correlated with kidney function markers, such as plasma creatinine and proteinuria.

Adiponectin is one of the most abundant adipokines in the circulation and has been identified as an exerkine, as several studies [26,27], including ours [12,19,20], have demonstrated changes in its circulating and tissue concentrations in response to exercise. Given its involvement in multiple metabolic processes, such as acting as an insulin sensitizer and promoting lipid beta-oxidation [12], adiponectin is a promising candidate for explaining, at least partially, the metabolic benefits of exercise [28]. However, the evidence in this area remains conflicting, as earlier studies have reported considerable variability in exercise’s acute and chronic effects on circulating adiponectin levels [29]. These differences between acute and chronic adaptations might be explained by the fact that acute responses to exercise could be associated with changes in the oligomerization of already secreted and/or pre-existing adiponectin pools, while chronic adaptations in adiponectin due to exercise might involve changes in both its production/secretion and oligomerization. This is particularly relevant given that both processes are highly sensitive to systemic metabolic states, such as redox balance and inflammation [30,31]. One limitation of many of these studies is their focus on total adiponectin levels, without distinguishing between its isoforms. Since adiponectin exists in different isoforms with varying levels of bioactivity [15], this limitation hampers a more comprehensive understanding of its role in exercise-induced metabolic changes. Adiponectin is predominantly found in circulation in three isoforms: low-molecular-weight (LMW), medium-molecular-weight (MMW), and high-molecular-weight (HMW), with bioactivity increasing across the isoforms [12,15]. Our results did not indicate a significant effect of a single maximal exercise session on LMW or MMW adiponectin levels, and we observed high inter-individual variability in the latter. This finding aligns with observations from other research groups [32]. Interestingly, the HMW isoform was undetectable in our sample, despite being recognized as the most biologically active form of adiponectin [33]. The formation of HMW adiponectin depends on several factors within the source cell, particularly in adipocytes. Studies have shown that a proper redox state in the endoplasmic reticulum [30] and a balanced environment of pro- and anti-inflammatory cytokines [34] are essential for the formation of HMW adiponectin [14]. In our sample, all participants were physically inactive, a condition known to promote redox imbalance and increase the production and secretion of pro-inflammatory cytokines [35]. This raises concerns, as the absence of the HMW isoform in a young adult population may reflect an early marker of increased cardiovascular risk, considering its established protective associations. It is important to highlight that in all participants we could not detect the HMW adiponectin isoform despite performing different experiments.

Exercise is widely recognized as an intervention capable of reducing circulating lipids (e.g., cholesterol and triglycerides) in populations at risk of cardiovascular diseases, such as those with obesity and type 2 diabetes [36]. However, high-intensity exercise can activate the sympathetic nervous system, particularly by increasing the production of catecholamines (i.e., adrenaline and noradrenaline) [37]. These catecholamines, in turn, are known to elevate circulating lipid levels [38]. This response is expected to be even more pronounced in individuals with low cardiorespiratory fitness, such as physically inactive adults [39], as observed in our sample. Moreover, given that the exercise protocol was performed to exhaustion, it is plausible to hypothesize that microscopic muscle damage may have released inflammatory cytokines into the circulation [40], thereby increasing cholesterol levels [41]. This potential muscle damage could also be associated with the acute increase in the albuminuria/creatinine clearance ratio [42] seen in our participants. The inflammatory cascade triggered by muscle damage could potentially affect both renal function and metabolic parameters simultaneously, offering an integrated explanation for our observations. Interestingly, the magnitude of change in total cholesterol was associated with changes in LMW adiponectin in our study. In this regard, it is debatable whether the increase in this isoform after high-intensity exercise is beneficial, given that it is the least bioactive form of adiponectin. Supporting this notion, Iwata et al. reported that, in a cohort of nearly 400 patients with type 2 diabetes, the LMW-to-total adiponectin ratio was higher compared to healthy controls and was positively associated with insulin resistance, as measured by HOMA-IR [43]. Moreover, in our study, the LMW adiponectin isoform was also positively associated with circulating FGF21 concentrations. Although FGF21 has biological functions similar to adiponectin [44], it has been shown that individuals with obesity may develop resistance to its actions [45]. Thus, increases in FGF21 concentration may not necessarily indicate a metabolic benefit. The mechanisms underlying FGF21 resistance remain unclear. However, several studies suggest that reduced expression of FGF receptors and its co-receptor β-klotho in target tissues (e.g., skeletal muscle and adipose tissue) may contribute to this phenomenon [46], an issue worthy of exploring in the future in some in vivo models.

In contrast to LMW adiponectin, MMW adiponectin exhibited negative associations with kidney function markers, such as circulating creatinine and proteinuria, suggesting that higher levels of this isoform may be associated with reduced kidney stress following maximal exercise. However, the literature on this topic presents conflicting results, primarily due to differences in the kidney function status of study participants. For example, a Korean study involving more than 2200 patients diagnosed with chronic kidney disease (CKD) found that higher adiponectin levels were negatively associated with the estimated glomerular filtration rate (eGFR) [47], indicating that elevated adiponectin could serve as a biomarker of renal dysfunction; however, only total adiponectin was measured. Interestingly, these findings are in agreement with what Moreno et al. described in patients with type 2 diabetes, where high serum adiponectin levels were inversely associated with eGFR even after controlling for several confounding factors such as sex, smoking habits, body mass index, waist circumference, diabetes duration, glycated hemoglobin (HbA1c), albumin creatinine ratio, and anti-hyperglycemic, anti-hypertensive, and anti-dyslipidemic treatments [48]. Conversely, Chen et al. observed a positive association between serum adiponectin levels and endothelial function, as measured by the vascular reactivity index (VRI), in a cohort of CKD patients who were not dialysis-dependent [49]. This suggests that the level of kidney function plays a critical role in determining the relationship between adiponectin and renal outcomes. The complexity of this relationship becomes even more apparent in individuals with proteinuria but without a CKD diagnosis. In this group, lower adiponectin levels were reported compared to healthy controls [50], highlighting the nuanced and context-dependent nature of adiponectin’s role in kidney function. We hypothesize that this complexity may, in part, stem from the differing bioactivity levels of adiponectin isoforms. In our sample, significant correlations with kidney function markers were observed exclusively with MMW adiponectin, while no such associations were found with the LMW isoform. However, we recognize the absence of studies focusing on the association of adiponectin isoforms with kidney function outcomes in young adults in the context of physical inactivity, particularly regarding acute changes as opposed to chronic alterations. Therefore, future studies should investigate the mechanisms underlying these differential associations to better understand the role of adiponectin isoforms in kidney function [51].

FGF21 has been proposed as a potential exerkine [52], given its previously reported changes following both acute [18] and chronic [22] exercise interventions. Interestingly, in our study, FGF21 levels did not change post-exercise, a finding that contrasts with previous reports. For instance, Slusher et al. observed a significant increase in circulating FGF21 following 30 min of continuous aerobic exercise in individuals with obesity [23]. However, key differences between studies may explain these discrepancies. Notably, the feeding status of participants in Slusher et al.’s study was not specified, whereas in our study, participants were in a fed state based on their habitual dietary patterns. Additionally, their study focused on individuals with obesity, while our participants were primarily classified as overweight. Furthermore, the exercise intensity in our study was higher. These factors are particularly relevant, as circulating FGF21 levels are influenced by both feeding state and exercise intensity [44]. Therefore, these variables should be carefully considered when evaluating the effects of exercise on FGF21 dynamics.

Among the limitations of this study, it is important to note that participants’ diets were not controlled, as they were instructed not to alter their usual food intake before the exercise test. Additionally, the inclusion of fasting levels for the metabolic outcomes of interest could have provided a deeper understanding of the participants’ baseline metabolic function, which will be incorporated in future studies. Another limitation of our study is the absence of additional post-exercise measurements (e.g., 1 h, 2 h, 4 h), despite the fact that the effects of a single exercise session can persist for several hours after its completion; therefore, future studies should consider implementing a more comprehensive temporal analysis to better understand the complete trajectory of these changes. Furthermore, the gender imbalance in our sample, with a higher proportion of women than men, may limit the generalizability of our findings. This issue will also be addressed in future research.

## 5. Conclusions

In conclusion, this study evaluated the metabolic and adiponectin isoform-specific responses to a single maximal exercise session in physically inactive young adults. While no significant changes were observed in LMW and MMW adiponectin levels, their associations with metabolic and kidney function markers suggest potential physiological relevance. LMW adiponectin was positively correlated with changes in total cholesterol and fibroblast growth factor 21 (FGF21), suggesting potential links to lipid metabolism and metabolic regulation. Conversely, MMW adiponectin showed negative correlations with kidney stress markers, including plasma creatinine and proteinuria, indicating a possible protective role in renal function under exercise-induced stress.

Interestingly, the low levels of high-molecular-weight (HMW) adiponectin underscore the potential impact of physical inactivity and metabolic states on adiponectin bioavailability, raising questions about its implications for metabolic and cardiovascular health in this population. These findings highlight the complexity of adiponectin’s role in exercise-induced metabolic responses and underscore the need for further research to elucidate the underlying mechanisms and clinical relevance of these associations [53]. Future studies should focus on the longitudinal effects of exercise, the role of adiponectin isoforms in different physiological and pathological states, and their potential as biomarkers for tailoring exercise interventions aimed at improving metabolic and renal health in physically inactive populations.

## Figures and Tables

**Figure 1 biomolecules-15-00314-f001:**
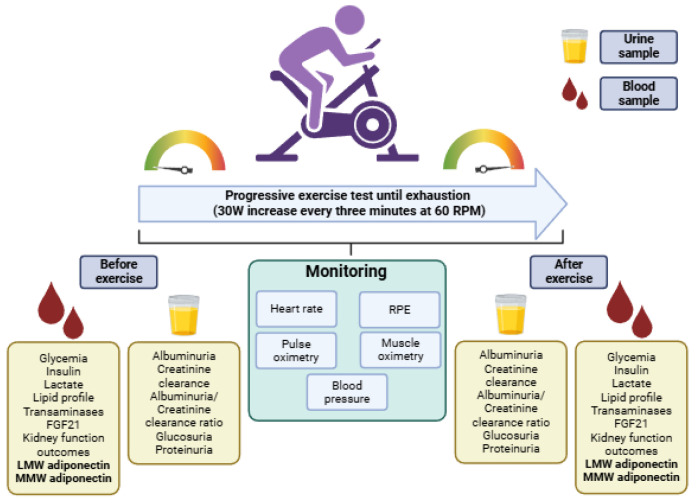
Study design, exercise session description, research outcomes, and time points. Abbreviations: LMW: low molecular weight; MMW: medium molecular weight; RPE: ratings of perceived exertion; FGF21: fibroblast growth factor 21; W: watts; RPM: revolutions per minute. Figure designed in Biorender.com.

**Figure 2 biomolecules-15-00314-f002:**
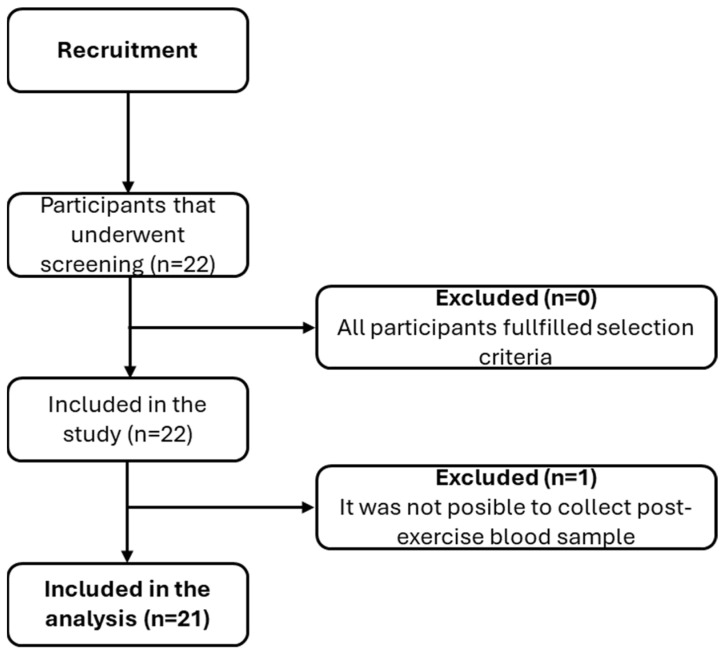
CONSORT model with the participants’ flow, selection criteria, and participants included in the study.

**Figure 3 biomolecules-15-00314-f003:**
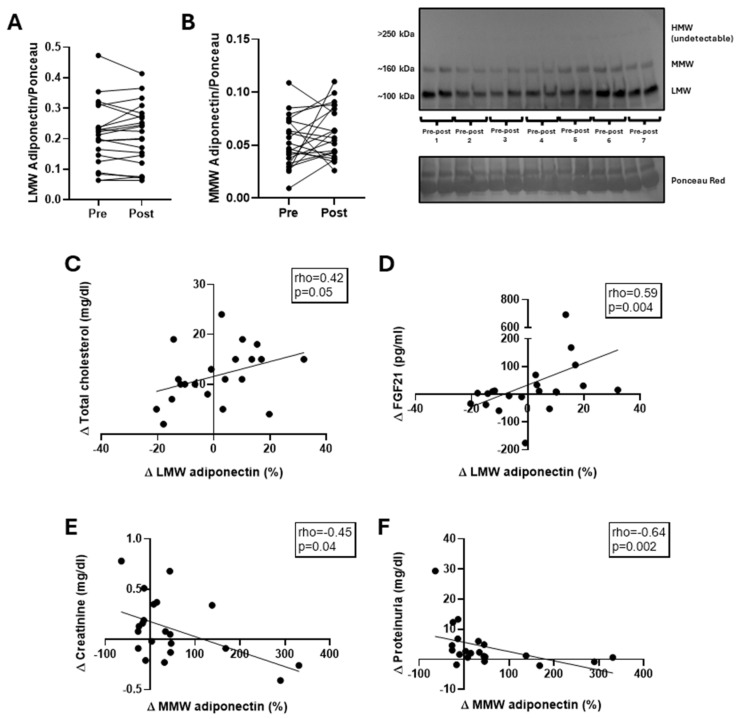
Effects of a maximal exercise session on adiponectin isoforms and its association with metabolic outcomes. (**A**) Low-molecular weight (LMW) adiponectin before and after the exercise session; (**B**) medium-molecular weight (MMW) adiponectin before and after the exercise session along with the representative blot with the adiponectin isoforms; (**C**) association between the changes (∆) of total cholesterol and the percentage of ∆ of LMW adiponectin before and after exercise; (**D**) association between the ∆ of the fibroblast growth factor (FGF)21 and the percentage of ∆ of LMW adiponectin before and after exercise; (**E**) association between the ∆ of plasma creatinine and the percentage of ∆ of MMW adiponectin before and after exercise; (**F**) association between the ∆ in proteinuria and the percentage of ∆ of MMW adiponectin before and after exercise. Original images of (**B**) can be found in Appendix A.

**Table 1 biomolecules-15-00314-t001:** Baseline anthropometric and physical activity characteristics of the participants.

Outcome	Frequency (x/x) or Median [IQR]
Sex (M/F)	3/18
Age (years)	24 [23–25.5]
Weight (kg)	66 [60–84]
Height (cm)	164 [158–167]
BMI (kg/m^2^)	25.6 [22.6–29.5]
Waist circumference (cm)	76 [73–87]
Hip circumference (cm)	102 [98–109]
Waist-to-hip ratio	0.75 [0.72–0.80]
Waist-to-height ratio	0.47 [0.45–0.55]
% Body fat	35 [28–41]
% Muscle mass	35 [32–39]
IPAQ (METs*min*week)	679 [330–775]
Siting time (hours)	5 [4–7.5]

Abbreviation list: BMI: body mass index; IPAQ: international physical activity questionnaire; MET: metabolic equivalent task. The values presented in the IPAQ section represent the total scores of this instrument, which are different from the self-reported physical activity used as inclusion criteria.

**Table 2 biomolecules-15-00314-t002:** Effects of one session of high-intensity interval training on cardiovascular and exercise-related outcomes.

Outcome	Before	After
SBP (mmHg)	121 [111–128]	126 [120–134] *
DBP (mmHg)	81 [73–86]	81 [76–85]
HR (beat/min)	85 [75–91]	184 [176–190] *
HR reserve used (%)	--	90 [82–93]
Recovery HR 3′ (beat/min)	--	47 [43–53]
Time test (min)	--	10 [10–12]
SatO_2_ (%)	99 [99–100]	99 [98–99]
SmO_2_ (%)	48 [41–53]	52 [40–63]
RPE (points)	6 [6–6]	17 [14–18] *
Lactate (mmol/L)	0.8 [0.8–1.8]	10.0 [6.1–11.6] *

Abbreviation list: SBP: systolic blood pressure; DBP: diastolic blood pressure; HR: heart rate; SatO_2_: pulse oximetry; SmO_2_: muscle oximetry. Values are presented as median [IQR], *: statistically different between before and after values (*p* < 0.05).

**Table 3 biomolecules-15-00314-t003:** Effects of one session of high-intensity interval training on circulatory metabolic outcomes.

Outcome	Before	After
Insulin (µUI/mL)	13.2 [8.7–27.2]	13.1 [10.3–21.2]
Glycemia (mg/dL)	88 [84–96]	93 [82–96]
FGF21 (pg/mL)	175 [121–415]	155 [111–503]
HOMA-IR	2.8 [1.8–5.3]	3.2 [2.3–4.6]
Total cholesterol (mg/dL)	151 [135–192]	162 [143–195] *
Triglycerides (mg/dL)	96 [70–135]	125 [70–147] *
HDL (mg/dL)	56 [51–67]	60 [55–74] *
LDL (mg/dL)	74 [62–102]	85 [65–109] *
VLDL (mg/dL)	19 [14–27]	25 [14–29]
No HDL (mg/dL)	95 [82–135]	104 [87–136] *
GOT (UI/L)	17 [14–25]	17 [14–27]
GPT (UI/L)	17 [10–24]	17 [10–24]

Abbreviation list: FGF21: fibroblast growth factor 21; HOMA-IR: homeostatic model assessment insulin resistance; HDL: high-density lipoprotein; LDL: low-density lipoprotein; VLDL: very low-density lipoprotein; GOT: glutamic-oxaloacetic transaminase; GPT: glutamate pyruvate transaminase. Values are presented as median [IQR], *: statistically different between before and after values (*p* < 0.05).

**Table 4 biomolecules-15-00314-t004:** Effects of one session of high-intensity interval training on kidney function outcomes (blood and urine).

Outcome	Before	After
Creatinine (mg/dL)	0.95 [0.81–1.04]	0.96 [0.78–1.26]
Uric acid (mg/dL)	3.7 [2.9–3.9]	3.6 [2.8–5.0]
Urea (mg/dL)	25 [21–31]	28 [22–30]
Blood urea nitrogen (mg/dL)	12 [10–14]	13 [10–14]
Estimated glomerular filtration rate (mL/min)	73 [70–90]	71 [56–89]
Albuminuria (mg/L)	0.7 [0.7–0.7]	0.7 [0.7–20.4] *
Creatinine clearance (g/L)	0.45 [0.29–0.85]	0.53 [0.21–0.78]
Albuminuria/creatinuria clearance ratio	1.8 [0.9–3.5]	2.2 [0.9–73.0] *
Glucosuria (mg/dL)	2.0 [0.7–3.7]	2.1 [0.6–4.2]
Proteinuria (mg/dL)	2.8 [1.9–4.5]	5.5 [2.2–10.1] *

Values are presented as median [IQR]. * Statistically different between before and after values (*p* < 0.05).

## Data Availability

Data are available upon request to the corresponding author at sergio.martinez@uss.cl.

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
