# Peer review of "Metabolic Response After a Single Maximal Exercise Session in Physically Inactive Young Adults (EASY Study): Relevancy of Adiponectin Isoforms"

_biomolecules, 2025, doi:10.3390/biom15030314_

Round 1

Reviewer 1 Report (Previous Reviewer 1)

Comments and Suggestions for Authors

The revised manuscript doesn't sufficiently address the comments. The major limitation is the lack of additional post-exercise time points.

Author Response

Dear reviewer,

we are grateful for your review and comments on our manuscript. Please find attached a reply point-by-point to every comment and suggestion made by you.

Kind regards.

Reviewer 2 Report (New Reviewer)

Comments and Suggestions for Authors

Abstract

Please spell out all abbreviations upon its first appearance, such as FGF21.

Statistical tests used to calculate P value should be briefly declared in the abstract.

Introduction

Please update reference 1. The provided web link is not functioning.

Line 71: Please explain what is meant by ‘our group’. Name of the group, purpose, nature, previous work of the group should be declared to enhance the reporting transparency.

Methods

Title of figure one should be detailed enough to allow understanding the nature and context of the study. Same comment applies for figure two.

Line 99: did you exclude those whom are physically active? If yes, please explain how the assessment was made to identify those whom are physically active prior to their exclusion.

Line 105: please explain why fibroblast growth factor 21 was used as a parameter to estimate the sample size and not another parameter? As explained in figure one, there are more than 18 variables being measured as per the protocol. Therefore, it is important for the reader to understand why FGF21 only was used for the estimation.

By consulting reference 23, it was not clear how a value of 50 pg/ml was selected as a parameter for the expected difference of the FGF21. Please explain.

Results

Title of tables should be revised to fully describe the context and nature of the study.

Table one: describing frequencies (such as male / female ) as Median and IQR is misleading. Please revise.

 Discussion

Line 292: a phrase seems missing. Please revise.

The discussion ignored the main parameter used to estimate the sample size. Please explain why the level of decreased during the comparison (175 vs. 155), although it was expected to increase as per the sample size estimation parameters (100 vs. 150).

Comments on the Quality of English Language

The manuscript can benefit from English language editing. 

Author Response

Dear reviewer,

we are grateful for your review and comments on our manuscript. Please find attached a reply point-by-point to every comment and suggestion made by you.

Kind regards.

Reviewer 3 Report (New Reviewer)

Comments and Suggestions for Authors

The manuscript titled “Metabolic response after a single maximal exercise session in physically inactive young adults (EASY-Study). Relevancy of adiponectin isoforms” presents a well-structured and insightful study that explores an important topic in exercise metabolism. The investigation into adiponectin isoforms and their role in metabolic and kidney function responses to acute maximal exercise is particularly relevant, given the increasing focus on exercise as a metabolic intervention for physically inactive populations.

The study is methodologically robust, with clear research objectives, a well-defined experimental design, and appropriate statistical analyses. The results are presented in an organized manner, and the discussion provides a thoughtful interpretation of the findings in the context of existing literature. Overall, the manuscript makes a strong contribution to the field of exercise physiology and metabolism.

While the manuscript is well-written and scientifically sound, there are a few minor areas for refinement that could enhance clarity and impact:

  1. Consistency in Terminology
    • The manuscript occasionally introduces abbreviations (e.g., LMW, MMW, HMW) without prior explanation. Defining them earlier in the text would improve readability.
    • The term "exercise-induced metabolic response" is used interchangeably with "metabolic adaptations to exercise". Standardizing terminology throughout the manuscript would enhance coherence.
  2. Refinement of Language and Sentence Structure
    • Some sentences are overly complex and could be streamlined for clarity.
    • Minor grammatical refinements would improve fluency, particularly in the discussion section, where certain arguments could be made more concise.
    • Furthermore, the authors coul cosidere the following recente article for the introduction: Messina et al., Functional effect of adiponectin and body composition assessment in lung cancer subjects after video-assisted thoracoscopic surgery (VATS) lobectomy, Thoracic Cancer, 2025, 16(2), e15260
  3. Future Research Directions
    • The discussion effectively highlights areas for future research, but a brief mention of the practical implications for exercise programming could be beneficial.
    • Since exercise intensity appears to influence adiponectin isoforms, discussing how different training protocols (e.g., interval training vs. continuous exercise) might impact these responses would add further value.

  1. Post-Exercise Follow-Up Measurements
    • The study focuses on immediate post-exercise changes, but including follow-up measurements (e.g., 1-hour, 4-hour post-exercise) in future research would help distinguish transient vs. sustained metabolic effects.

Author Response

Dear reviewer,

we are grateful for your review and comments on our manuscript. Please find attached a reply point-by-point to every comment and suggestion made by you.

Kind regards.

Round 2

Reviewer 1 Report (Previous Reviewer 1)

Comments and Suggestions for Authors

no further comments

This manuscript is a resubmission of an earlier submission. The following is a list of the peer review reports and author responses from that submission.

Round 1

Reviewer 1 Report

Comments and Suggestions for Authors

The present study investigates the effect of a short time exercise intervention on primarily adipnectin at different molecular weight. Moreover different meatbolic marker and kidney markers are measured before and after exercise. Although no significant alteration of adiponectin level are measured between pre and post exercise correlation of delta adiponectin (pre-post) and delta metabolic and kidney markers are given. The study shows a sex imbalance which could influence the results. The study has also further limitations given in the specific comments.

Specific comments:

1.      It should be discussed that acute and chronic adaptation of adiponection level in the blood must be interpreted differential. Acute exercise reflect an acute elease of already storaged adiponectins while chronic changes reflect mainly an alteration in adiponectin production.

2.      Is there a correlation between the time up to exhaustion and the measured adiponectin concentration.

3.      Is there an alteration of the findings if only women data are analyzed?

4.      A shortcoming of the given study is the lack of a further post-exercise measurement. An additional measurement e.g. 1h after exercise could give more information about possible mechanistic relation between adiponectin and the other measured parameter.

5.      It is questionable if the correlation of the parameters give information about the interaction of adiponectin with metabolic and kidney function.

6.      It seems difficult in the discussion that correlation between chronic alteration of metabolic factors and kidney function are compared with the acute changes investigated in the given study.

Reviewer 2 Report

Comments and Suggestions for Authors

The study investigates the effects of a progressive exercise session on circulating adiponectin isoforms and their associations with metabolic and kidney stress markers in physically inactive young adults. The study particularly addresses isoform-specific adiponectin levels which is a limitation in the literature. This is a well-conducted study that contributes to understanding adiponectin’s role in exercise physiology. However, some areas that could be improved to enhance the manuscript’s clarity:

·       Since non-parametric tests were used, it suggests that the data are not normally distributed. In such cases, it would be more appropriate to report the median and interquartile range (IQR) instead of means and standard deviations.

·       The participants selection criteria in the methods section states “less than 150 minutes/week of moderate physical activity and/or 75 minutes/week of vigorous physical activity.” However, in the baseline table, the data reported are unclear, does it reflect total physical activity? Please clarify and ensure the data used for participant selection are reported.

·       Additionally, the possibility that intense exercise caused microscopic muscle damage and inflammation should be considered. This could lead to the release of inflammatory molecules that affect cholesterol metabolism, potentially explaining the temporary increase in total cholesterol levels. This hypothesis aligns with the observed increase in the albuminuria/creatinine clearance ratio and could complement the catecholamine explanation provided in the manuscript. this mechanism might also explain the association with adiponectin, which could be secreted by skeletal muscles in response to exercise. Including this discussion would strengthen the interpretation of the findings.

Reviewer 3 Report

Comments and Suggestions for Authors

1. The author reported HMW was undetectable. However, the HMW is one of the most biologically active form of adiponectin. Does the assay not work? or the level is below the detection range?

2. The author only tested the metabolic response right before and after the exercise session.  The changes and effects that could happen later and the knowledge the readers could get should be claimed and discussed.

3. Some writing should be improved, like p should be italicized, space should be added between words and brackets.